# A Dynamic Four-Step Data Security Model for Data in Cloud Computing Based on Cryptography and Steganography

**DOI:** 10.3390/s22031109

**Published:** 2022-02-01

**Authors:** Rose Adee, Haralambos Mouratidis

**Affiliations:** 1Department of Computer and Systems Sciences, Stockholm University, Borgarfjordsgatan 12, Kista, 164 40 Stockholm, Sweden; road0872@student.su.se; 2Institute for Analytics and Data Science, University of Essex, Colchester CO4 3SQ, UK

**Keywords:** cybersecurity, cloud computing, cryptography, steganography, security model, data, privacy

## Abstract

Cloud computing is a rapidly expanding field. It allows users to access computer system resources as needed, particularly data storage and computational power, without managing them directly. This paper aims to create a data security model based on cryptography and steganography for data in cloud computing that seeks to reduce existing security and privacy concerns, such as data loss, data manipulation, and data theft. To identify the problem and determine its core cause, we studied various literature on existing cloud computing security models. This study utilizes design science research methodology. The design science research approach includes problem identification, requirements elicitation, artifact design and development, demonstration, and assessment. Design thinking and the Python programming language are used to build the artifact, and discussion about its working is represented using histograms, tables, and algorithms. This paper’s output is a four-step data security model based on Rivest–Shamir–Adleman, Advanced Encryption Standard, and identity-based encryption algorithms alongside Least Significant Bit steganography. The four steps are data protection and security through encryption algorithms, steganography, data backup and recovery, and data sharing. This proposed approach ensures more cloud data redundancy, flexibility, efficiency, and security by protecting data confidentiality, privacy, and integrity from attackers.

## 1. Introduction

Cloud computing technology has been growing exponentially [1]. More and more companies choose cloud computing services since they are flexible, reliable, scalable, and in most cases, the most affordable solution [1]. Cloud computing is defined as an emerging and popular method of accessing shared and dynamically configurable resources via the computer network on demand [2]. An example of a cloud computing service today includes Amazon Elastic Compute Cloud (Amazon EC2), which supports virtual information technology (virtual IT) and allows users to rent virtual computers to run their computer applications [3]. Amazon EC2 provides scalable computing capacity in the Amazon Web Services (AWS) Cloud [4]. Other examples of cloud computing services include the Google App Engine used for application hosting. The Google Apps and Microsoft Office Online are examples of software as a service, the Apple iCloud is used for network storage, and DigitalOcean is used for servers, which serves as both an infrastructure and a platform as a service [3].

### 1.1. Service Models of Cloud Computing

There are three main service models of cloud computing—Infrastructure as a Service (IaaS), Platform as a Service (PaaS), and Software as a Service (SaaS).

IaaS. Infrastructure as a Service (IaaS) is a computing service that is hugely scalable and automated [5]. Computers, networking, storage, and other resources can all be accessed and monitored using IaaS. Instead of purchasing hardware outright, IaaS helps companies purchase services online and as-needed [5].

PaaS. Platform as a Service (PaaS) provides the cloud components of specific apps and is mainly used for applications [5]. PaaS provides developers with a platform to create and configure applications. The enterprise or a third-party vendor can handle all servers, storage, and networking, while the developers can manage the applications [5].

SaaS. Software as a Service is the most widely used solution for companies in the cloud industry [5]. It is a cloud application service, also known as cloud application hosting. SaaS uses the internet to distribute services to customers that a third-party provider runs. Most SaaS applications run directly in the web browser, so no client-side downloads or installs are needed [5].

### 1.2. Cryptography

Cryptography is a practice that involves the study of secure communication techniques to prevent unauthorized third parties from accessing private data, information, or messages [6]. The practice incorporates various aspects of the information security field, such as data confidentiality, integrity, and authentication (CIA) and non-repudiation, the focal areas in modern cryptography [7].

Encryption for the cloud computing world is an important issue requiring investigation in several studies [8]. Jaber and Bin argue that an example of a significant focus area of encryption in cloud computing is identification based on encryption [9]. Since cloud computing manages crucial data and is accessible anywhere globally through the internet, security is a critical factor and significant concern [10]. Cryptography can play a crucial role in secure data transmission, e-commerce, digital media privacy, and web data storage and transmission [11]. Several algorithms are fitted for encryption and decryption to ensure data security in cloud computing; these include Data Encryption Standard (DES), Advanced Encryption Standard (AES), Identity Based Encryption (IBE), and the Rivest, Shamir, Adleman, Algorithm (RSA) [10].

### 1.3. Symmetric and Asymmetric Encryption Method

These methods’ main difference is that asymmetric encryption uses a pair of public and private keys to encrypt and decrypt messages while transmitting. In contrast, symmetric encryption uses a single key shared by those who wish to access the message [12]. Asymmetric encryption is a comparatively modern strategy compared to symmetric encryption [12]. Therefore, asymmetric encryption was created to solve the underlying issue of exchanging the key in symmetric encryption models by using a pair of public-private keys to remove the need to share the key. However, when compared to symmetric encryption, asymmetric encryption is slower [12].

### 1.4. AES (Advanced Encryption Standard)

AES (Advanced Encryption Standard) is a symmetric block cipher that is a method of encrypting text. AES uses plain text in blocks of 128 bits and, by using keys of 128, 192, and 256, converts them into ciphertext [7]. The number of transformation rounds that translate the input, known as plaintext, into the final output, known as ciphertext, is specified by the key size used by an AES cipher [11]. Each round includes multiple processing stages, one of which is dependent on the encryption key. Then, a series of reverse rounds convert ciphertext to plaintext [11]. The downside of AES is that it uses a simple algebraic structure. Also, it is hard to implement [7].

### 1.5. RSA (Rivest–Shamir–Adleman)

RSA (Rivest–Shamir–Adleman) is one of the most popular encryption and decryption algorithms. RSA encryption is frequently used with several other encryption schemes and digital signatures that prove a message’s validity and legitimacy [10]. However, since it is less reliable and resource-intensive than symmetric-key encryption, it is not commonly used to encrypt entire messages or data [10]. Thus, developers use other encryption schemes combined with RSA to enhance security and efficiency [7]. Only an entity with access to the RSA private key will decrypt the symmetric key using this method [7]. Without the symmetric key, message decryption is almost impossible. Therefore, combining two or more encryption and decryption algorithms is an elegant solution for enhancing security and efficiency; moreover, it is a standard solution [7].

### 1.6. IBE (Identity Based Encryption)

Identity-based encryption, also known as ID-based encryption, is a form of public-key encryption (PKE) that uses an identifier as the encryption mechanism’s foundation [13]. Parties of an IBE communication will encrypt messages (or check signatures) without exchanging keys, which is useful when key sharing is infeasible or technically impossible [13]. A private key and its identification (ID) are used to compute a public key using an identity-ID, for example, an email [13]. Then, it will send encrypted messages to the person/entity associated with the identity-ID using this computed public key.

### 1.7. LSB (Least Significant Bit)

The Least Significant Bit (LSB) in steganography is the technique where the least significant bit of an image (a byte consists of 8 bits and the least significant bit number 8) is replaced with a bit of data [14]. Steganography is the practice of hiding information within another block of information or a physical object [14]. As technology strives forward, several tools and innovations utilize old steganographic methods such as null ciphers, picture coding, audio, and video [14].

When those data bits are combined, they form a secret message. After hiding the secret message, the cover image is almost similar to the stego-image under consideration [14]. However, specific image characteristics are almost impossible to be detected by the naked eye; image steganography takes advantage of that embedded data and avoids detection [1]. However, as this method is vulnerable to steganalysis, we encrypt the raw data before embedding it in the image [14]. Although the encryption method adds to the complexity of the process, it also adds to the security. LSB embedding is also applied in specific data domains, such as embedding a secret message in the RGB bitmap data’s color values or the frequency coefficients of a JPEG picture. LSB embedding can be used in a wide range of data formats and types. As a result, LSB embedding is one of the most widely used steganography methods today [14].

The model will use steganography and cryptography to add an extra layer of security. Moreover, this method, combined with a cryptographic algorithm, has been proven to be an effective solution in other practices [7]. Thus, we have chosen RSA in combination with AES to take advantage of the speed of symmetric encryption and the security of asymmetric encryption and LSB as a steganographic method to enhance security by combining them.

The LSB algorithm would be used to hide the encrypted AES key in a cover object, thus enhancing security even further. Since the AES key would be encrypted and hidden in an image or a file, the sender’s encrypted data would be sent to the receiver. Then the receiver would find the encrypted key hidden in the cover file and, finally, decrypt it using the RSA private key.

### 1.8. Research Problem

The general problem stems from cloud computing being increasingly adopted by many organizations. As a result, some security and privacy concerns, such as data manipulation, data loss, and theft, arise from using any device to load and retrieve data from the cloud providers’ facilities [10]. One of the serious issues that may arise is unauthorized access by the insiders [2]. Although there are many techniques to prevent unauthorized access by the cloud administrators, the techniques provided to prevent the unauthorized access by the cloud administrators to the client data in the cloud have proved insufficient [15].

Unauthorized access to frequently sensitive client data may result in data breaches where sensitive data leak out, causing problems for most organizations that have adopted cloud computing [16]. Consequently, the effects of data leakage through cyberattacks and eavesdropping may be detrimental to the organizations that employ cloud computing services. The effects include loss of data and leakage of sensitive and confidential information, which leads to loss of clients’ trust in the companies and can also contribute to substantial financial setbacks [14].

This paper addresses the problem of insecurities and privacy breaches of some existing cloud computing security models that employ encryption algorithms in data storage and transmission. Encryption by itself has proved to be insufficient in providing data security and maintaining privacy for data in storage and transmission in the cloud computing environment [7]. A computer science security model refers to a scheme specifying and imposing security policies [17]. A security model can be built on top of a structured model of access privileges, a computation model, a model of distributed computing, or no particular theoretical grounding at all [17].

The problem is of general interest and great significance because it affects many people on an individual and business or organizational level, cutting across many industries, especially in the IT industry and academia [18]. In this paper, the stakeholders affected by the security and privacy breaches on cloud computing data include the organizations that employ cloud computing services, customers or clients of those organizations, employees, individuals, IT systems administrators, and third-party cloud service providers.

An example of the existing models includes the three-step data security model based on cryptography and steganography [7]. Another example is the cryptographic role-based access control model for electronic health record (EHR) systems. The EHR system uses location-and biometrics-based user authentication and a steganography-based technique to embed EHR data in electrocardiography (ECG) host signals [19]. Likewise, the PHR (Personal Health Record) model, which medical staff use in exchanging patients’ health information, is another example of existing models [20]. A final example is the private cloud for software as a Service (SaaS) application model [6].

Both the private cloud for software as a service model (SaaS) [6] and the three-step data security model based on cryptography and steganography [7] do not facilitate data backups before the data is transmitted. Therefore, there is a need to introduce and develop a data security model with an added security layer to back up the data and employ steganography alongside the encryption algorithms used in cloud computing to improve the data’s security and privacy [21].

The model proposed by this paper is perceived to be original in using cryptographic algorithms such as RSA and AES together with IBE and LSB steganographic techniques in a four-step model. Many existing data security models for cloud computing mainly employ cryptographic algorithms. However, very few security models employ cryptography and steganography [7]. In addition, none of the other security models reviewed provide a concrete solution for secure data sharing. In contrast, the proposed security model employs the IBE technique to facilitate secure data sharing.

### 1.9. Research Question

This paper aims to answer the following question: How can we enhance data security and maintain privacy in cloud computing environments?

### 1.10. Research Goal and Objectives

This paper’s research goal is to enhance data security and maintain privacy in cloud computing environments by reducing the number of security and privacy issues in cloud computing, for example, data loss, data manipulation, and data theft. This study therefore aims to achieve the following objectives.

To investigate how cryptography applies in the cloud computing security model;To investigate how to apply steganography alongside cryptography in the cloud computing security model.

The proposed artifact can classify as a data security model that uses four steps. Data protection and security through encryption algorithms is employed in the first stage, steganography is employed in the second stage, data backup and recovery are performed in the third stage, and data sharing is performed in the fourth stage. Therefore, the proposed model uses the four steps to provide an extra layer of security to preserve data confidentiality, integrity, and cloud data availability. The model proposed by this paper is perceived to be original in using cryptographic algorithms such as RSA and AES together with IBE and LSB steganographic techniques in a four-step model.

## 2. Materials and Methods

### 2.1. Design Science Research Methodology

The paper implements the cloud data security model using design science research methodology (DSRM). The design science research (DSR) process includes five steps or activities [22], namely: Problem explication (defining the research problem and justifying the value of a solution); requirements elicitation (the definition of objectives for a solution); artifact design and development (the design and development of artifacts such as constructs, models, and methods); demonstration (the demonstration by using the artifact to solve the problem); and evaluation (comparing the objectives and the actual observed results from the use of the artifact and communication of the problem). Table A1 in Appendix A indicates the design science research canvas.

### 2.2. Explicate the Problem

In this phase, we review different literature on existing cloud computing security models to define the problem and ascertain the root cause of the research problem.

### 2.3. Related Work

An author, Ghuge, proposed an application model to help implement a high security level in SaaS applications [6]. According to Ghuge, Software as a Service (SaaS) is one such service that delivers services to the end-users in a pay-as-you-go manner. Nevertheless, security often seems to be a significant drawback despite SaaS’s advantages [6]. Therefore, the paper proposed an application model for any SaaS application hosted on a private cloud environment. The application divides the services into two micro-services. The first one is an application layer firewall [6]. The second is a secure application to log in and send sensitive data. The application layer firewall checks for any malicious activity and prevents the intruder from accessing the application’s features. Subsequently, they implement a hidden Markov model layer, a probability-based intrusion detection technique [6]. The second micro-service uses the Advanced Encryption Standard (AES) encryption algorithm to encrypt documents with sensitive data transmission within the private cloud. Further security is provided by proposing a novel video steganography approach using the Least Significant Bit (LSB) technique [6]. Therefore, the paper provides a detailed structure of hiding the data using multiple security levels [6].

Another author says that cloud computing provides pay-as-you-go services through the internet [1]. In this paradigm, steganography and cryptography are some of the technologies utilized in the cloud to protect user data transmission. A review of various studies was conducted in this work, with a focus on the Least Significant Bit (LSB) and Discrete Cosine Transform (DCT) approaches [1]. In one of the reviews, a publication described a steganography technique that provides multi-level security by segmenting the secret text message into numerous cells and randomized each segment at the bit level before embedding the characters of the secret text message into a cover picture using 2D-DCT [1]. This randomization or modification is accomplished utilizing a one-of-a-kind function with a reversal. All of the text parts are put into various places of the cover picture at random. To make the number of characters in each unit equal, a bit-stuffing approach is utilized. A new pseudo-random sequence generator function is used to build a pseudo-random sequence to embed each of the units of the secret message into the logical square sections or blocks of the cover picture in a pseudo-random method [1]. Extraction of the original message may be done using the same pseudo-random sequence. As previously stated, the secret password and a passkey known only to the two intended parties (sender and receiver) determine the input value designated threshold of the pseudo-random sequence generator function [1]. Since they are secret values, an additional level of security is applied.

The article’s authors utilize a hybrid encryption technique rather than a single encryption algorithm to secure cloud storage [23]. The hybrid method of AES and FHE is the authors’ main focus. In contrast to previous approaches, this hybrid technique allows the user to maintain data, and is more redundant and safer. To produce hybrid encryption calculations based on AES and RSA in Bluetooth innovation, the authors took use of the procedure speed of secret key encryption and the suitable key administration of common key encryption [23]. They believe that if AES can use a 256-piece square figure with 14 cycles for encryption, this innovation may also be used in cloud computing. The encryption procedure in the second phase is based on a fully homomorphic encryption method. This approach accomplishes two goals: additional substance and multiplicative homomorphic [23]. The user will only use the substance calculations that have been added. The user has ciphertext from the maiden scramble and is using the private key at this point. Using added substance homomorphic encryption, the Cipher content and secret key will now be encoded together [23]. The user can secure data confidentiality, privacy, and integrity from hackers by employing this strategy. Users may learn more about how this strategy works by looking at the flow chart and algorithm in the methodology section of this article [23].

The authors of the study suggested that encryption aids in the transmission of sensitive data via an unsecured channel without the risk of data loss or manipulation by an unauthorized party [24]. Asymmetric encryption, commonly known as public-key encryption or holomorphic encryption, is the subject of this study. Asymmetric encryption, on the other hand, is usually utilized for key exchange rather than data encryption due to the enormous key size. According to the authors, data security is a major concern in today’s massive data centers and cloud computing [24]. Elliptic Curve Cryptography is used in this study to encrypt data in the cloud since the key used in Elliptic Curve Cryptography is comparatively small. Elliptic Curve’s computing power is minimized due to its tiny key size, resulting in the least amount of energy usage [24]. ECC is utilized for encryption, key creation, and decryption in this work. Point P (x, y) selection is critical in building a safe and dependable encryption method. This study proposes a two-layered solution to data security in the cloud. The first step is to divide the data into small chunks, and the second is to encrypt it with random safe curves. The two stages will secure data security to the point where a quantum computer system may not be able to breach it [24]. For data encryption, the dynamic Elliptic Curve technology has been used.

According to the authors of this article, combining cryptosystems with steganography has been found as an upgraded security paradigm for data transmission [25]. To satisfy cloud-specific communication efficiency, it will need to be improved at both phases. Using Hybrid Cryptosystems and an Adaptive Genetic Algorithm aided by the Least Significant Bit (LSB) embedding procedure as inspiration, an effective Visually Imperceptible Hybrid Crypto Steganography (VIHCS) model is proposed in this article [25]. The authors created a unique Hybrid Cryptosystem by carefully combining AES and Rivest–Shamir–Adleman (RSA) algorithms to protect secret data that are then inserted in a cover picture. Furthermore, the usage of the AGA-OPAP (Adaptive Genetic Algorithm-based Optimal Pixel Adjustment) enhanced the Least Significant Bit embedding while maintaining the highest possible picture quality and ocular imperceptibility [25]. The authors used the 2D-Discrete Wavelet Transform (2D-DWT-2L) approach with 8×8-dimensional block-wise embedding to accomplish LSB embedding. When used in conjunction with the AGA-OPAP model, it aids in improving embedding efficiency [25].

According to the authors, advances in the sphere of information technology are requiring us to safeguard the privacy of digital data [26]. Combining cryptography with steganography is among the most effective techniques to attain such concealment. A unique RGB shuffling algorithm is proposed in this study [26]. The idea behind RGB shuffling encryption is to shuffle all of the RGB elements in order to distort the image. The RGB shuffling technique shuffles the RGB values of each pixel in a picture based on the password entered by the user [26]. Adding an RGB element with an ASCII password, inverting it, and shuffling it is the first stage in RGB shuffling. The message is encrypted in this study using the Message Digest 5 (MD5) Algorithm, and the authors employ RGB shuffling to encrypt the picture. They then used Least Significant Bit (LSB) methods to embed the encrypted information in an image, video, or audio [26]. The information or file is hidden in the rightmost bit using the LSB technique.

### 2.4. Overview of the Related Data Security Models

Several articles regarding existing cloud computing security models are reviewed in the document review process and the information obtained is represented in Table 1 below.

### 2.5. Define Requirements

To define the security model’s requirements, we use existing data security models for cloud computing. A requirement is a property of an artifact that is needed, wanted, or desired by stakeholders in practice. It may guide the design and development of the artifact [22]. This paper’s requirements definition was done through a document review concerning cloud security models as the primary source of knowledge. Through the document review, we were able to identify both functional and non-functional requirements of the security model. In addition to document reviews, we also employ survey questionnaires in this phase to validate and provide justification for some of the functional and non-functional requirements of the security model obtained from the document reviews.

The following functional and non-functional requirements were deemed necessary to ensure the study’s proposed security model was functional and could be used to address the research problem. The problem regards providing a model to enhance data security and maintain privacy in cloud computing environments. Each requirement was either found in similar projects or extracted and adapted to our model from similar projects. The discussion section justifies the requirements based on the questionnaire responses.

#### 2.5.1. Functional Requirements

A functional requirement in the proposed security model explains what the security model is supposed to do for the users and what benefit it provides [22]. The functional requirements are discussed below.

The model shall improve data security through encryption and decryption. The idea of the model is to perform cryptography; namely, encryption and decryption of cloud data that are adapted from the cryptographic role-based access control model for electronic health record (EHR) systems. The EHR, an electronic health security model, uses a cryptographic role-based technique to distribute session keys to establish communications and information retrieval using the Kerberos protocol [19]. Furthermore, the use of cryptography as a requirement is also adapted from the three-step data security model that applies cryptography using an RSA algorithm in its first step [7]. Also, cryptography is borrowed from other articles that explore the use of cryptography in cloud computing. Two examples are this article on hybrid Schnorr, RSA, and AES cryptosystem [27] and this article on the comparative analysis of DES, AES, and RSA crypt algorithms for network security in cloud computing [28].

Rationale: This paper’s proposed model uses cryptography to generate a hidden key that encrypts data using the AES and RSA encryption algorithms. The receiver receives both the encoded letter and the secret key for decryption.

2.The model shall add an extra layer of security by applying steganography. The idea of applying steganography as a functional requirement in the security model was adapted from reviewing the three-step data security model [7]. The three-step data security model enlists applying steganography to hide data within an image in its second step [7]. Furthermore, the idea of using steganography was reinforced by reviewing research on image encryption based on AES and RSA algorithms [29]. Likewise, a review of data security in cloud computing using steganography in which the authors apply the LSB technique further supports this idea [1].

Rationale: The idea was that after the first phase of encrypting the data using cryptography, an extra layer of security would be added using the Least Significant Bit (LSB) steganography technique. LSB is the method of modifying the last bit of a block of bytes in a file [14]. Furthermore, when the last bits are merged, they form bytes, which are combined to form information.

3.The model shall provide data backups. The requirement for the security model to offer the capability for backing up data is adapted from the security model to enhance mobile cloud computing security using steganography [21]. The authors of the model that works with a key embedded in the image and the data to provide an additional security layer for data confidentiality argue that the users are responsible for downloading and/or uploading information from or to the cloud and creating backup files [21]. We conducted more research on disaster recovery techniques in cloud computing [30] and cloud-based disaster recovery and planning models [31].

Rationale: This is a method of making duplicate backups of data and saving them somewhere to retrieve the original in the event of a data failure. Full backups, incremental backups, or other backup forms may be used based on the user’s preferences.

4.The model shall enable data recovery in the cloud. The idea of using data recovery as a requirement in the proposed security model is also adapted from the security model to enhance mobile cloud computing security using steganography [21]. The authors of the security model suggest that data backup and recovery processes should be implemented as one technique in the security model [21]. Similarly, we researched disaster recovery techniques in cloud computing [30] and cloud-based disaster recovery and planning models [31]. The idea of using data recovery as a requirement of the security model was thus strengthened.

Rationale: The idea was that the model would make it easier to recover data that are unavailable, misplaced, corrupted, destroyed, or formatted from secondary storage; portable media; or files that could not be accessed commonly.

5.The model shall facilitate secure data sharing. The idea of the security model to facilitate secure data sharing is adapted from the PHR (Personal Health Record) security model for exchanging patients’ health information [20]. The PHR service enables the patient data to be securely stored in a third-party server so that authorized persons can share and check the PHR data of the patient [20]. Further research indicates that secure data sharing in cloud computing is essential [32,33].

Rationale: The security model will make cloud file sharing easier. The mechanism in which a person is given storage space on a computer and reads and writes are done over the internet is known as cloud-based file sharing or online file sharing. In most cases, the administrator can delegate access rights to other users if they see fit.

#### 2.5.2. Non-Functional Requirements

A non-functional requirement specifies in what way or how well the security model should provide its functions [22]. The non-functional requirements are discussed as follows.

The model shall be simple. The idea to have simplicity as a non-functional requirement of the security model is adapted from an application model to help implement a high security level in SaaS applications [6]. The author of the application model argues that the LSB method is picked because it is a fundamental and straightforward method for concealing data and can be easily understood by end-users [6].

Rationale: A security model’s primary goal is to effectively provide the degree of understanding required to execute crucial security specifications. Thus, the model shall be easy to use and learn.

2.The model shall be reliable. The idea of using reliability as a non-functional requirement in the security model is derived from the three-step data security model [7]. The authors of the three-step model argue that the three steps should be reliable to secure the data from outsiders or hackers [7]. Furthermore, the review of another paper discussing disaster recovery techniques in cloud computing reinforced the idea of using reliability as a non-functional requirement of the security model [30].

Rationale: The model shall be reliable regarding data availability in the cloud. Furthermore, one of the main features of cloud computing, availability, is a crucial feature that our model should provide [7]. Since the model would be handling data in the cloud, availability is essential.

3.The model shall be scalable. The scalability requirement was formed and adapted from the cryptographic role-based access control model for electronic health record (EHR) systems. The EHR system uses location- and biometrics-based user authentication and a steganography-based technique to embed EHR data in electrocardiography (ECG) host signals [19]. The authors of the EHR model argue that the model shall be scalable due to the increasing number of users [19]. Similarly, the authors of disaster recovery techniques in cloud computing argue in favor of scalability as a requirement of cloud security models [30].

Rationale: The model shall handle and perform well under an increased and expanding workload or scope. Because of today’s system architecture complexities, many information management models cannot keep up with the changing market and technological climate. For the model to be an elegant choice for security in cloud computing, it shall guarantee its ability to secure data in the cloud even when the environment it in which it is applied is expanding.

4.The model shall be effective. The idea for the security model to have effectiveness as a non-functional requirement was adapted from the model that uses a three-level defense system structure. Each floor performs its duty to ensure the data security of cloud layers [2]. Moreover, the model’s authors argue that its effectiveness in cloud computing and its efficiency are crucial to successful data protection [2]. Likewise, the paper’s authors, who discuss cloud-based disaster recovery and planning models, argue that cloud-based security models should provide practical solutions to replace legacy disaster recovery strategies [31].

Rationale: The model shall guarantee security goals such as integrity and data availability in cloud computing. Moreover, since we are creating a security model, the primary goals to be met are data integrity, availability, and confidentiality [7].

5.The model shall be ethical. This requirement derives from the three-step data security model [7]. The authors argue that the model shall provide confidentiality, since it is one of the most vital security aspects [7]. Furthermore, ethicality derives from the model’s ability to provide data confidentiality and security. Data confidentiality is also adapted from the paper discussing security concerns and countermeasures in cloud computing [34].

Rationale: The model shall adhere to ethical norms by promoting security goals such as confidentiality. Since we are in an era where correct data handling is one of the most critical issues, our model shall handle data with respect and careful adherence to the data handling procedure.

## 3. Results

### 3.1. Design and Develop the Artifact

During the design and construction process, we use a design thinking approach. Empathizing, defining, ideating, prototyping, and evaluating the solution are design thinking approaches we use. For the data collection process, we perform document reviews.

This paper’s artifact is classified as a model according to design science research. The paper’s output is a four-step data security model for cloud computing based on encryption algorithms and steganography. The four steps that the security model contains are described as follows.

#### 3.1.1. Step 1: Data Security and Privacy through Encryption

In the first step, cryptography was used in the security model to enhance data security and maintain privacy in the cloud. To take advantage of the speed of symmetric encryption and the security of asymmetric encryption and steganographic methods, we propose an algorithm that combines AES and RSA.

AES, as a symmetric algorithm, requires the same key in both encryption and decryption. As a result, the big problem of disseminating a hidden key to hundreds of people without fear of compromise occurs. An elegant solution to this matter is the combination of AES and RSA encryption [7]. The benefits of combining the two algorithms are that there is no need for a mutual secret. The sender only needs to know the recipient’s public key, which lowers the possibility of leakage [7].

The AES algorithm encrypts and decrypts large data objects, as it is faster. AES is used to encrypt sent data, which exploits its high encryption speed and low RAM requirements. Only the key uses the slower RSA algorithm. RSA protects the encryption key from theft by generating two keys (private and public). Media Access Control (MAC) protects the encrypted private key or the data. Even if an attacker obtains the data, the data are still fully protected since the attacker must have access to the private key to decrypt the data [6]. Lastly, this method is easily expandable if there is a need to distribute the same data to multiple recipients, using multiple copies of the encrypted AES key and a separate public key [7].

##### How AES and RSA Work in the Security Model

The process starts with a sender and a receiver, where the sender wants to securely share files with the receiver. The sender generates an RSA key pair which includes a private and a public key. A trusted third party is then used to distribute the private key to the receiver through key management. The key pair distribution is followed by generating an AES key. Then, the AES key is used to encrypt and later decrypt the plain text. The RSA public key is then used to encrypt the ciphertext and the AES key to guarantee safety. Together with the ciphertext, the encrypted AES key is sent to the receiver, where the receiver decrypts the AES key and data using their private RSA key. Therefore, the result is the initial plain text that was encrypted.

The steps of the combined algorithms of the security model are:The user generates an RSA main pair.The sender generates an AES256 key at random. The AES256 key is a one-time usage key.The AES key is used to encrypt the files.The RSA public key is used to encrypt the AES key and the ciphertext.The receiver receives the encrypted data as well as the encrypted key.The receiver then uses their RSA private key to decrypt the AES key.The data are decrypted by the receiver using the AES key.

The description and steps for how AES and RSA work in the model are illustrated in Figure 1 below.

#### 3.1.2. Step 2: Applying Steganography

The second step involves adding an extra layer of security to the data hidden using steganography. This section explains how the Least Significant Bit (LSB) steganography technique is applied in the security model in the second step and its rationale.

The LSB method is very straightforward. Many techniques for hiding messages inside multimedia carrier data are based on the LSB embedding method [14]. LSB embedding is also applied in specific data domains, such as embedding a secret message in the RGB bitmap data’s color values or the frequency coefficients of a JPEG picture. LSB embedding can be used in a wide range of data formats and types. As a result, LSB embedding is one of the most widely used steganography methods today [14].

As denoted in our requirements section, our model will use steganography and cryptography to add an extra layer of security. Moreover, this method, combined with a cryptographic algorithm, has been proven to be an effective solution in other practices [7]. Thus, we have chosen RSA in combination with AES to take advantage of the speed of symmetric encryption and the security of asymmetric encryption and LSB as a steganographic method to enhance security by combining them.

The LSB algorithm would be used to hide the encrypted AES key in a cover object, thus enhancing security even further. Since the AES key would be encrypted and hidden in an image or a file, the sender’s encrypted data would be sent to the receiver. Then, the receiver would find the encrypted key hidden in the cover file and, finally, decrypt it using the RSA private key.

##### How the LSB Steganographic Technique Works in the Security Model

The process starts with a sender and a receiver, where the sender wants to securely share files with the receiver. The sender generates an RSA key pair which includes a private and a public key. A trusted third party is then used to distribute the private key to the receiver through key management. The key pair distribution is followed by generating an AES key. Then, the AES key is used to encrypt and later decrypt the plain text. The RSA public key is then used to encrypt the ciphertext and the AES key to guarantee safety. The LSB algorithm then hides the encrypted AES key and ciphertext in an image. The stego cipher image is then sent to the receiver, where the receiver first has to extract the ciphertext and key from the stego cipher image before decrypting it.

Therefore, the steps using the LSB algorithm in the security model are:The user generates an RSA main pair.The sender generates an AES256 key at random. The AES256 key is a one-time usage key.The AES key is used to encrypt the files.The RSA public key is used to encrypt the AES key and the ciphertext.The encrypted AES key and ciphertext are hidden using the LSB algorithm in an image.The receiver receives the stego ciphertext image.The receiver extracts the ciphertext from the stego cipher imageThe receiver uses their RSA private key to decrypt the AES key.The data are decrypted by the receiver using the AES key.

The steps for using the LSB algorithm in the second phase of the model are illustrated in Figure 2 below.

#### 3.1.3. Step 3: Perform Data Backup and Data Recovery

The third step describes how the security model performs data backups and recovery. This section also explains the backup approaches supported by the security model. The security model employs three backup approaches: full backups, incremental backups, and differential backups.

Compared to conventional approaches such as disks and tapes, where handling and shipping media tend to be challenging activities, cloud storage for backing up data by businesses is economical and cost-efficient [35]. On the other hand, the move to cloud computing poses challenges to factors critical to a company’s growth, such as availability and security [35].

Data backup and disaster recovery use a range of approaches, depending on the needs of the business, such as:Full backup: data are backed up on a full scale and restored on a full scale [35].Incremental backup: only changed or newly added data are backed up subsequently after the last full or incremental backup. The last full and incremental backups are done every day from the last full backup used to restore the data [35].Differential backup: only modified or newly inserted data since the last complete or differential backup are backed up for a differential backup. However, the previous differential backup adjustments are updated in the differential backup [35], simplifying the recovery process. Therefore, it only includes the most recent complete backup and differential backup copies.

The needs of the enterprise determine data backup and disaster recovery strategies. These arrangements differ depending on the size of the operation and the volume of data to be backed up. Our model will not enforce a single technique since every organization has different objectives, goals, scales, and priorities for data backup techniques. Any popular and proven method can be adapted as a part of our four-step security model.

##### How Backup and Recovery Work in the Security Model

The process starts with a sender and a receiver, where the sender wants to securely share files with the receiver. The sender generates an RSA key pair which includes a private and a public key. A trusted third party is then used to distribute the private key to the receiver through key management. The key pair distribution is followed by generating an AES key. Then, the AES key is used to encrypt and later decrypt the plain text. The RSA public key is then used to encrypt the ciphertext and the AES key to guarantee safety. The LSB algorithm then hides the encrypted AES key and ciphertext in an image. The stego cipher image is backed up using any backup technique and approach supported by the security model. The stego cipher image is then sent to the receiver, where the receiver proceeds to extract the ciphertext and key from the stego cipher image before decrypting it.

Therefore, the steps for backup and recovery in the security model are:The user generates an RSA main pair.The sender generates an AES256 key at random. The AES256 key is a one-time usage key.The AES key is used to encrypt the files.The RSA public key is used to encrypt the AES key and the ciphertext.The encrypted AES key and ciphertext are hidden using the LSB algorithm in an image.The stego cipher image is backed up by the chosen technique or approach.The receiver receives the stego ciphertext image.The receiver extracts the ciphertext from the stego cipher imageThe receiver uses their RSA private key to decrypt the AES key.The data are decrypted by the receiver using the AES key.

Figure 3 below illustrates how the data backup step of the security model is performed.

#### 3.1.4. Step 4: Perform Data Sharing

In the fourth step, the security model performs secure data sharing. The security model uses RSA, AES, and identity-based encryption (IBE) to securely transfer data to authorized persons. This section explains how the security model uses IBE to securely share data.

Data sharing aims to give data owners the right to delegate different access rights to their data to other cloud users [13]. The cloud should be ready to support varying needs so that data owners may grant or remove access privileges to other users, allowing them to edit their data [13]. Additionally, users’ privacy must be shielded against the cloud, allowing them to hide their personal information when using the cloud [36]. Finally, users should access shared data in the cloud through connected devices with limited processing power, such as smartphones and tablets [37].

While cloud storage appeals to customers and businesses since it allows for large-scale data sharing, it does not guarantee users’ privacy or data protection [37]. Despite the ease of using cloud file-sharing systems, users must rely on the service provider’s ability to deliver high availability and timely backup and recovery [38]. If company data are hosted on third-party providers without the IT department’s awareness, cloud file storage can pose security and compliance issues to the enterprise. Users’ data are exposed to various risks and malicious attacks resulting from providing complete access to cloud resources, and security breaches are common [38]. For example, certain clouds may be untrustworthy with data secrecy for financial reasons; sensitive information may be leaked to market competitors or the cloud service provider may hide data loss to protect their credibility [39].

One of the steps in this paper’s proposed model aims to provide a reliable cloud data storage service that enables users to have dynamic access to their data. When data owners outsource their data to the cloud, they expect a high degree of encryption and confidentiality [37]. Although users typically encrypt their data while storing it on a cloud server, they want control over it, for example, if they regularly change it.

An elegant solution to satisfy this aim is to employ a data sharing security technique that utilizes the RSA and AES algorithms combined with the identity-based encryption (IBE) technique [13].

##### How Secure Data Share Works in the Security Model

In the first step of the proposed model, encryption is applied using AES and RSA. For data share, this step is enhanced with the additional help of the IBE technique. Furthermore, the secure sharing of cloud data is satisfied by users obtaining their unique private keys.

The process starts with a sender and a receiver, where the sender wants to securely share files with the receiver. The sender generates an RSA key pair which includes a private and a public key. A trusted third party is then used to distribute the private key to the receiver through key management. The key pair distribution is followed by generating an AES key. Then, the AES key is used to encrypt and later decrypt the plain text. The RSA public key is then used to encrypt the ciphertext and the AES key to guarantee safety. The LSB algorithm then hides the encrypted AES key and ciphertext in an image. The stego cipher image is backed up using any backup technique and approach supported by the security model. The stego cipher image is then sent to the receiver, where the receiver proceeds to extract the ciphertext and key from the stego cipher image before decrypting it.

Therefore, the steps for backup and recovery in the security model are:The user generates an RSA main pair.The sender generates an AES256 key at random. The AES256 key is a one-time usage key.The AES key is used to encrypt the files.The RSA public key is used to encrypt the AES key and the ciphertext.The encrypted AES key and ciphertext are hidden using the LSB algorithm in an image.The stego cipher image is backed up by the chosen technique or approach.The receiver receives the stego ciphertext image.The receiver extracts the ciphertext from the stego cipher imageThe receiver uses their RSA private key to decrypt the AES key.The data are decrypted by the receiver using the AES key.

Figure 4 below illustrates how data share is performed using the security model and contains a summary of all the steps of the model.

### 3.2. Demonstrate and Evaluate the Artifact

For evaluating the artifact, we use an artificial strategy. An artificial evaluation refers to evaluating an artifact in an artificial setting, for example, a laboratory, using toy problems and non-practitioners [22]. Thus, the evaluation strategy we embrace is known as an ex-ante evaluation. An ex-ante evaluation means that the artifact is evaluated before the event and without being used [22].

We evaluated the security model in terms of access time and the processing speed rate of the AES and RSA algorithms. The Python programming language was utilized to implement the proposed scheme and test its efficiency. Before encrypting and decrypting, we had to install Python libraries that would enable us to use all of the helper encryption and decryption algorithms that we desired to use. Some of the libraries installed using pip included NumPy, AES, RSA, PIL, CV, Cryptodome, and Matplotlib.

Our message was “Rose Adee encrypted files”. we had to hold the message in a global variable message, from whence we could access and encrypt it using AES, RSA, and later on, LSB steganography to embed and hide our encrypted message into the image.

We chose to first encrypt the message with AES encryption; the main Python libraries used were AES and Cryptodome. We used the AES encryption process to create ciphertext, which is an unreadable, effectively indecipherable conversion of our message. The output of the encryption process, the AES ciphertext, could not be read until the secret AES key generated using the AES library was used to decrypt it.

The AES uses a block size of 128 bit = 16 bytes, so to sustain optimal encryption performance, we provide at least 10/16 × 10^^9^ encryptions per second. In our model, the AES-256, which has a key length of 256 bits, is used. The AES-256 supports the largest bit size and is practically unbreakable by brute force based on current computing power, making it a strong encryption standard [25].

After the initial round, the process is repeated 9, 11, or 13 times, depending on whether the AES algorithm is using a key length of 128 bits, 192 bits, or 256 bits. 128-bit AES encryption undergoes 10 transformation rounds; 192-bit AES encryption undergoes 12 transformation rounds; and 256-bit AES encryption undergoes 14 transformation rounds. Since the AES algorithm only uses one secret key to cipher and decipher information, it requires less computational power than RSA, making it faster and more efficient to run [26]. Since our message was only a few lines of text, the process took 0.121765 s to encrypt and decrypt our message, which is quite fast.

Continuing with our encryption, we passed our bigger ciphered message through RSA encryption as well. With RSA, we encrypted sensitive information (our message) with a public key, and a matching private key is used to decrypt the encrypted message.

The RSA is considerably less efficient, more resource-heavy, and slow due to its calculations with large numbers. In particular, the decryption where d is used in the exponent is slow. There are ways to speed it up by remembering p and q, but it is still slow in comparison to the AES algorithm [25]. The security of the RSA cryptosystem is based on the problem of factoring large integer numbers. This is because the equation n = p*q holds, where p and q are primes. n is the common part of the private and public keypair. If you know the public keypair (n, e), it is possible to also calculate the private keypair (n, d), because d is simply the multiplicative inverse of e modulo (p − 1) * (q − 1) [25]. This is the RSA key generation formula:(M e ) d (mod n) = M, n = pq

In our Python script, we used libraries like RSA, crypto, and PKCS1_OAEP, which is a hybrid of the AES and RSA algorithms. However, when we tried to run our now large file through RSA encryption, we ran into an error that claimed our message was too long. After extensive research, we found that the solution was to use a hybrid algorithm that involved the use of PKCSI_OAEP encryption. Then, our ciphered text was successfully ciphered further. After timing the process, we found that RSA encryption took longer, perhaps due to the fact that our message was now larger. It took a 1.186813 s to encrypt and decrypt the message.

Since increasing protection levels requires some time spent on encryption and decryption, our security model therefore leverages the speed of the AES symmetric algorithm and the security of the RSA asymmetric algorithm in a hybrid combination of AES and RSA. The combination also helps to solve the problem of disseminating hidden keys to hundreds of people without fear of compromise [23].

After further encrypting our message, we decided to hide its content within three different images of varying sizes and color scales. Among the images was one of 1.2 MB, another of 2.9 MB, and the largest of 7.2 MB. For this, we used LSB steganography, an image steganography technique in which messages are hidden inside an image by replacing each pixel’s least significant bit with the bits of the message to be hidden. Each pixel contains three values, which are Red, Green, Blue. These values range from 0 to 255; in other words, they are 8-bit values. We go further to draw histograms of the encrypted and unencrypted images to study the different distortions to RGB pixels plotted against the value of these pixels. The libraries we used to achieve this include PIL and NumPy. Unlike cryptography, which conceals the contents of a secret message, steganography conceals the very fact that a message is communicated. However, according to our observations in this activity, grayscale images seemed to be slightly different from their unencrypted versions.

Table 2 below describes the response time in seconds using the different encryption algorithms.

The cover photos, stego images, and their histograms are shown in Table 3. When observing the histograms with bare eyes, the findings demonstrate that they are identical. As a result, the degree of distortion in the stego picture is minimal.

### 3.3. Application of Informed Arguments to Evaluate the Artifact

We also utilize a descriptive approach to employing informed arguments using information from the knowledge base [22]. We employed informed arguments to provide arguments and deliberations indicating how our model fares in comparison to the requirements and techniques used in some of the related works.

Similarly, to the papers [6,25,26], the proposed security model applies a combination of cryptographic algorithms and steganography techniques. The proposed model applies a new hybrid combination of AES-256, RSA, and IBE encryption together with LSB steganography to fulfill the aim of the study, which was to enhance the security and privacy of cloud data. In the proposed model, encrypting the data multiple times does not necessarily increase its security, but applying the AES-256 keys along with LSB steganography makes brute-force attacks on the cloud data twice as long to perform, therefore increasing protection of the data.

In relation to papers [1,6,24,25], there is no clear mention of data backups and recovery procedures. Paper [23], on the other hand, maintains data redundancy and the security of its data, while paper [26], mentions image recovery in one of the phases. In the proposed scheme, cloud data redundancy offers an extra layer of protection and reinforces the backup by replicating data using one of the users’ chosen backup techniques to an additional system. We employ cloud services like Google Drive to back up the bank’s data with the user. Moreover, Google Drive, by default, supports incremental backups but can also perform full backups of the user files [35], which increases reliability in terms of availability. Cloud data redundancy is necessary as it ensures that in the event of disturbances to the cloud operations, there are fallbacks within the cloud architecture and the businesses can continue operating as expected.

In the papers [1,6], data sharing and transmission happens through delivering the services in a pay-as-you-go manner. In paper [23], data transmission is through Bluetooth, while paper [25] uses combined cryptosystems with steganography for data transmission. Papers [24,27] have no specific mention of how data are shared using those models. However, in the proposed scheme, the AES-256 keys are all encrypted before data sharing is performed. With the proposed scheme, cloud administrators can delegate access rights to the users as they see fit. Additionally, unlike the other papers in the related works, the proposed scheme facilitates secure data sharing by employing identity-based encryption (IBE). With the fast advancement of cloud computing, an increasing number of people and businesses are storing and exchanging data on the public cloud. Therefore, to preserve the privacy of data stored in the cloud, a data owner normally encrypts it so that only specific authorized users may decode it. The notion is supported by different researchers who argue that cloud-based file sharing can be more secure by employing cryptographic algorithms [20].

## 4. Conclusions

The authors have been able to effectively integrate cryptography and steganography security approaches to give twice the protection for cloud data security and privacy. To protect cloud data, we introduced a dynamic four-step model with hybrid encryption, in which the AES-256 symmetric method is paired with the RSA asymmetric technique. The encrypted data are then concealed in a photo using the LSB steganography technique. The users’ chosen strategies can be used to back up the results of the decryption process. With identity-based encryption (IBE), the results of the encryption and decryption may be shared and securely transferred to authorized recipients. The results also reveal that when the picture distortion is minimized, the quantity of data concealed in the image rises. For diverse companies of varied sizes, objectives, and demands, the suggested methodology is more flexible, adaptable, and efficient for safeguarding cloud data. In comparison to other comparable efforts, the approach additionally assures cloud data redundancy. The qualities of the proposed model make it suited for data exchange in the cloud, financial, and healthcare environments. The model can safeguard the confidentiality, privacy, and integrity of cloud data by employing the approaches described. As a result, since the model verifies data integrity, it can be concluded that the goal of this work, which was to improve data security and the privacy of cloud data, has been met. The security goals of looking at how cryptography and steganography are used are also met. Nonetheless, more research on how to improve the combination and provide greater security for multimedia data is necessary in the future.

## Figures and Tables

**Figure 1 sensors-22-01109-f001:**
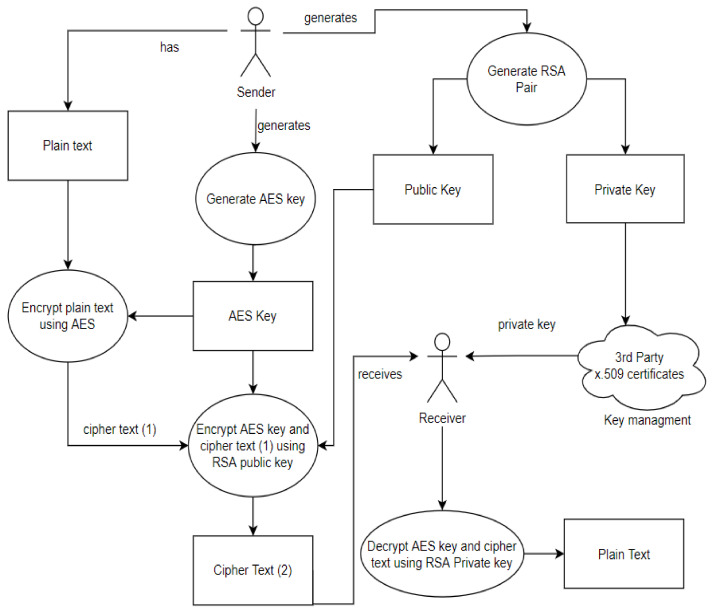
A description of how AES and RSA work in the first step.

**Figure 2 sensors-22-01109-f002:**
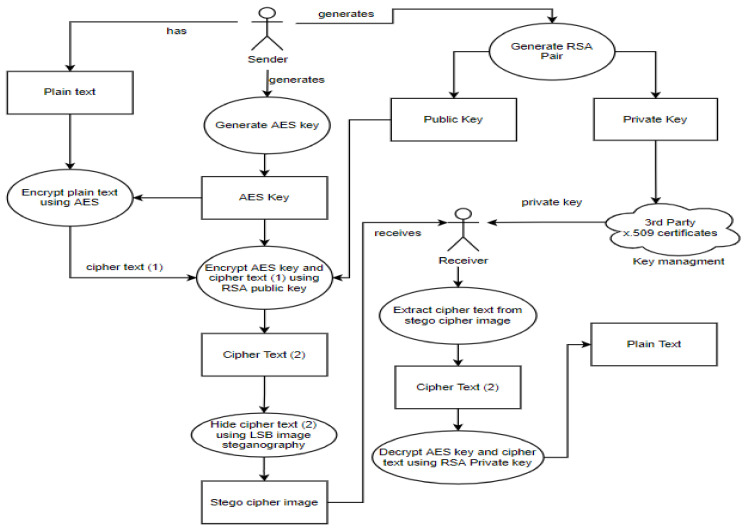
A description of how the LSB steganography works in the security model.

**Figure 3 sensors-22-01109-f003:**
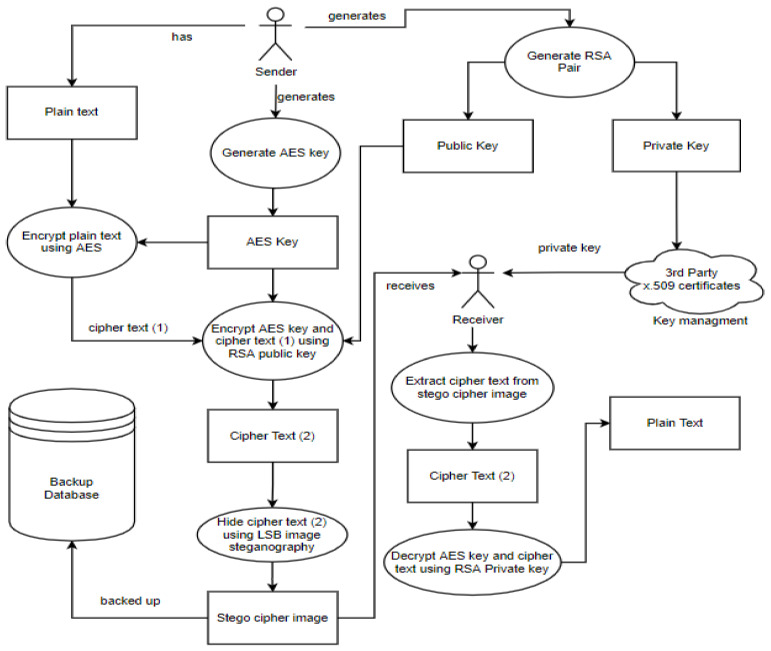
A description of the data backup step of the security model.

**Figure 4 sensors-22-01109-f004:**
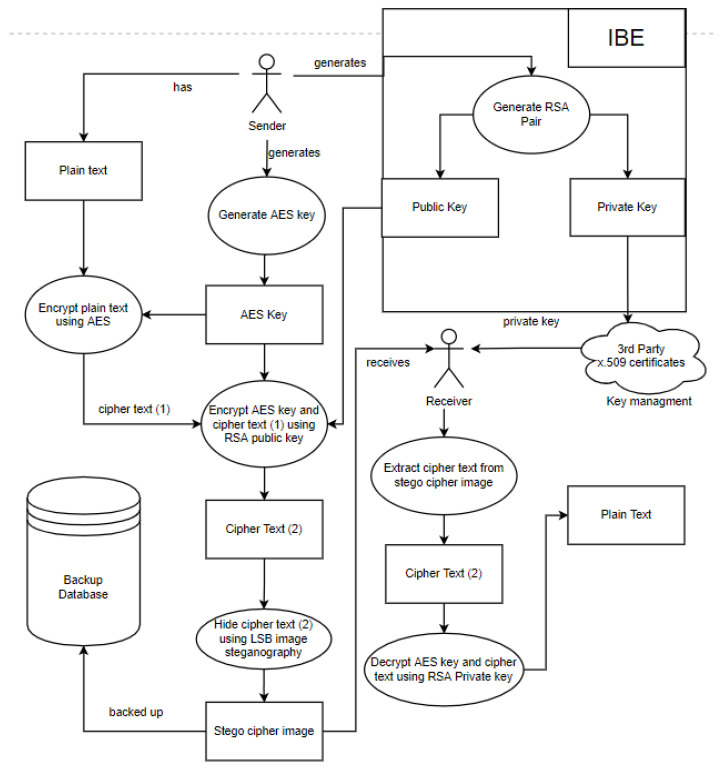
A description of the four steps of the security model.

**Table 1 sensors-22-01109-t001:** The table below summarizes the security models reviewed in the document review process.

Security Model	Cryptographic Algorithms	Steganography Technique	Backup and Recovery	Data Share	References
The private cloud for software as a service (SaaS)	AES	LSB video technique of steganography	No mention of data backups and recovery	Deliver services to end users in a pay-as-you-go manner	[6]
Image-Based Steganography Using Pseudorandom Sequence Generator Function and DCT Coefficients	No clear mention of cryptographic algorithms	LSB Image steganography using pseudo-random sequence function with 2D-DCT	No mention of data backups and recovery	Provide services in a pay-as-you-go manner	[1]
The hybrid encryption in Bluetooth innovation and in cloud computing	AES, FHE	Steganography not applied	Maintain data redundancy and security	Bluetooth	[23]
Data security in cloud computing using Elliptic Curve Cryptography	ECC	Steganography not applied	No mention of backups and recovery	Data share not specified	[24]
Visually Imperceptible Hybrid Crypto Steganography (VIHCS) model	AES, RSA	2D-Discrete Wavelet Transform (2D-DWT-2L) AGA-OPAP with LSB	No clear mention of backups and recovery	Combined cryptosystems with Steganography for data transmission	[25]
RGB shuffling method using combined steganography and cryptography	RGB shuffling algorithm and Message Digest 5 (MD5) algorithm	LSB image, video, or audio technique of steganography	Mention of image recovery in one of the phases	No specific mention of data sharing using the model	[26]

**Table 2 sensors-22-01109-t002:** The table shows the response time in seconds.

Response Time in Seconds							
Cover image	Size	LSB Encryption	LSB decryption	Total LSB time			
1	1.2 MB	6.618632	0.681449	7.300081			
2	2.9 MB	12.194479	1.134509	13.328988			
3	7.2 MB	31.071637	2.842029	33.913666			
**Message Encryption**							
AES Encryption time	AES decryption time		Total AES time	RSA Encryption time	RSA decryption time	Total RSA time	Total time
0.011895	0.002815		0.121765	0.502411	0.684402	1.186813	1.308578

**Table 3 sensors-22-01109-t003:** The table below shows the cover images, stego images, and the histograms.

Cover1 image	Histogram of cover1 image
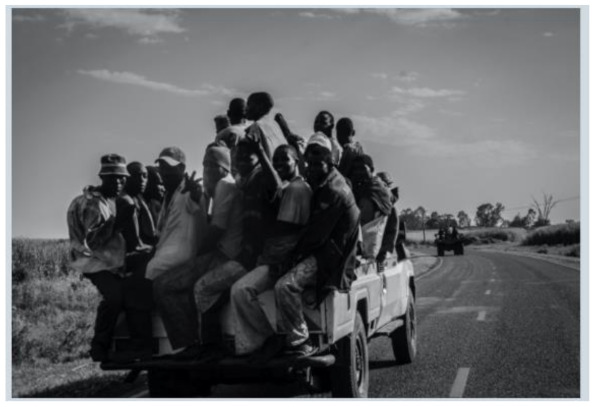	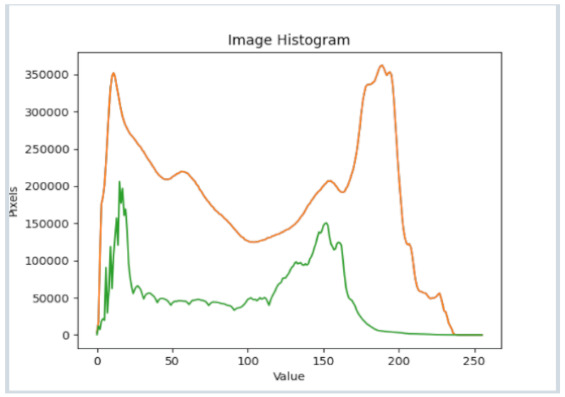
Stego1 image	Histogram of Stego1 image
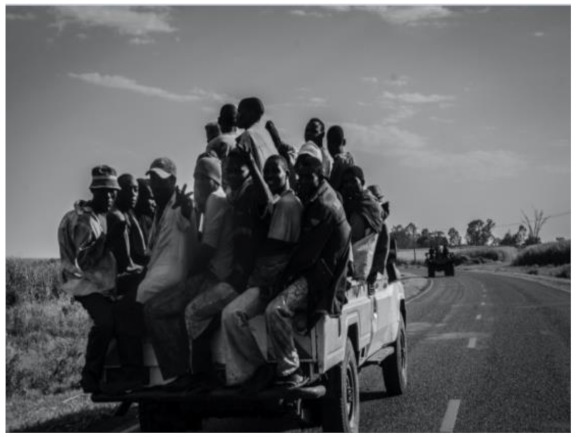	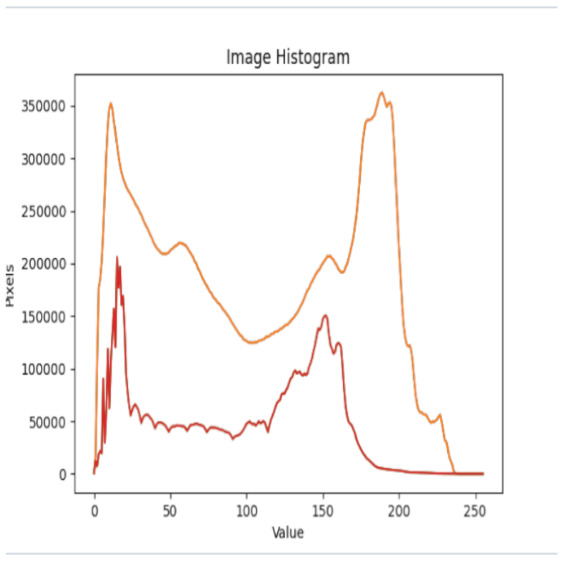
Cover2 image	Histogram of cover2 image
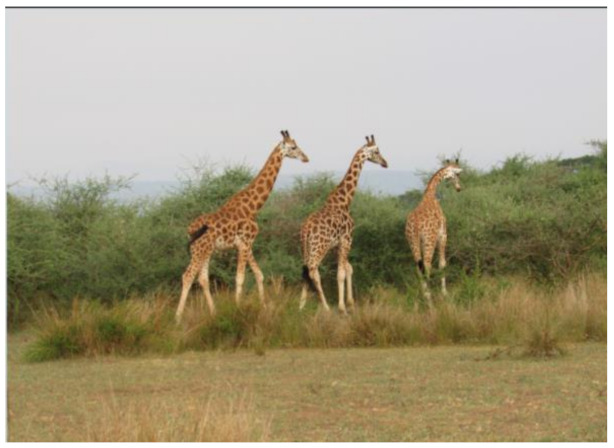	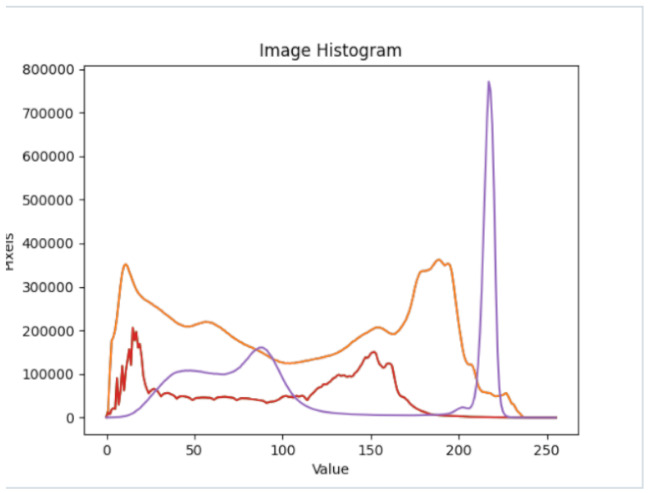
Stego2 image	Histogram of Stego2 image
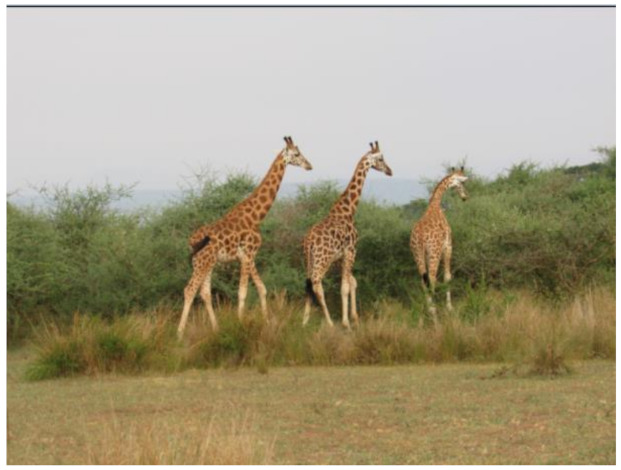	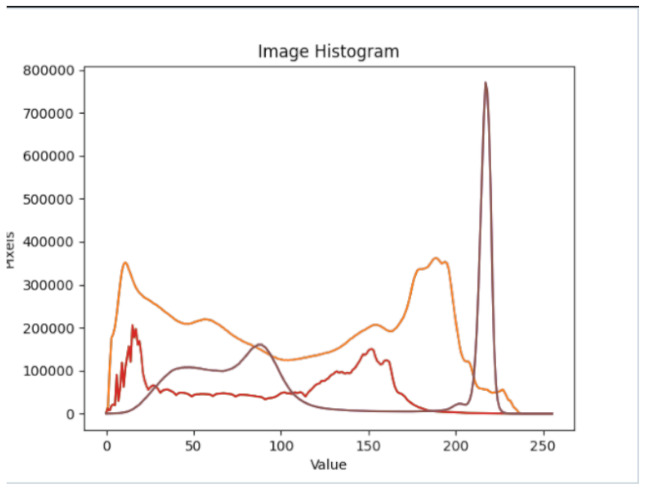
Cover3 image	Histogram of cover3 image
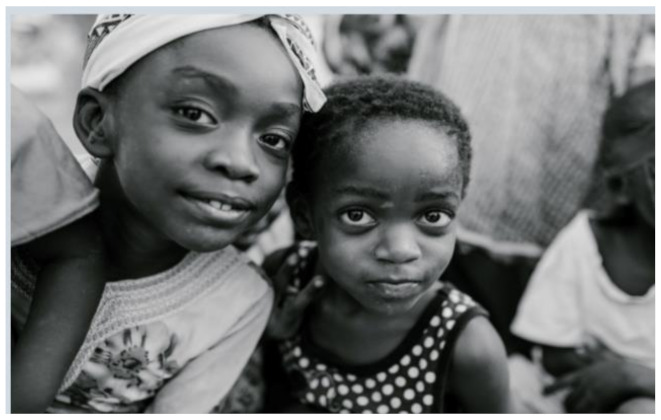	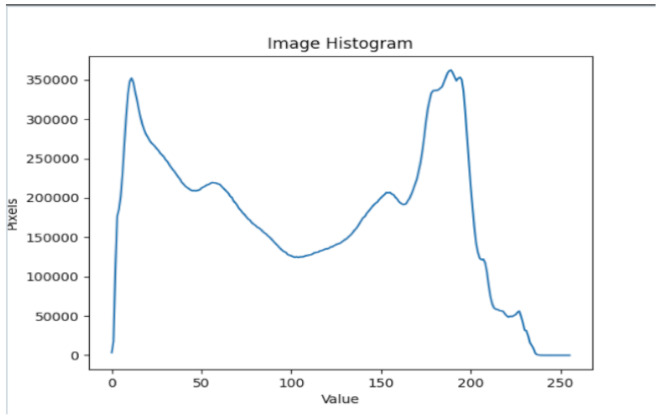
Stego3 image	Histogram of stego3 image
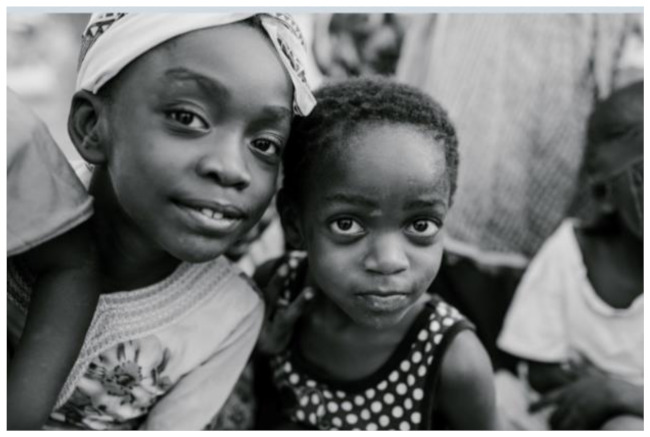	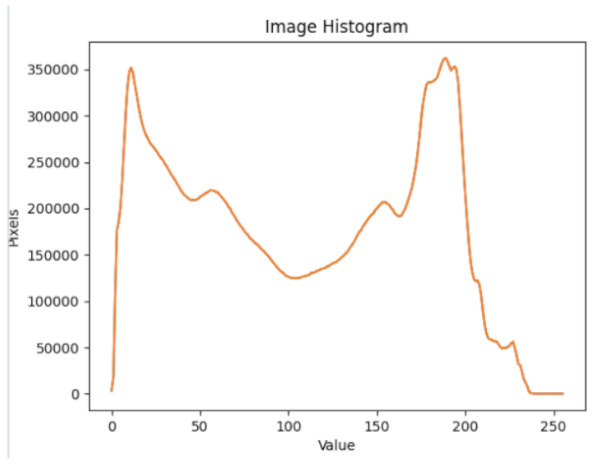

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
