# Peer review of "A Dynamic Four-Step Data Security Model for Data in Cloud Computing Based on Cryptography and Steganography"

_sensors, 2022, doi:10.3390/s22031109_

Round 1

Reviewer 1 Report

The purpose of the paper is to study the possibility of cryptography for data protection in cloud computing. In Intro the authors give a good overview of data security models, which is a rewriting of well-known papers in other words, and do not contribute new material to understanding the problem. New information does not follow from this, and this part can be reduced to a couple of paragraphs without problems to the understanding of the paper. It is not necessary to draw this in detail. The paper from 26 pages will be three times shorter and save the reader time.

The authors have not left the paradigm of using a combination of known methods of protection by simply superposition. If public-key cryptography is not enough, let's encrypt the data twice by steganography as well. So you can increase protection but why it is necessary? There is no proof that using just plain encryption is not enough. The authors describe their proposed model of protection well but do not provide mathematical proof or give an example of the need
for protective approaches. I guess that the material will be better if it is done.

The authors do not evaluate the availability of the proposed approach in access time or processing speed rate in the cloud computing process. It is necessary because increasing the protection levels requires some time spent on encryption and decryption.

The language of the manuscript is clear without grammatical problems. Taking into account the comments above, the paper may be published.

Reviewer 2 Report

This article applies a dynamic four-step data security model to data in cloud computing.  Here are some reminders and suggestions to authors.

Reviewer 3 Report

The topic of the article is to ensure data security in the cloud through the use of cryptography and steganography. The authors propose a 4-step model using known algorithms such as AES or the LSB method.

The topic of the article is interesting and up-to-date. However, I find the style in which it is written inappropriately. The article is way too long. The authors describe each step very precisely, and thus it is difficult to grasp the essence of this article. In addition, it resembles a project report (e.g., figures showing how to generate a key in a given algorithm). Besides, I have doubts about the novelty of the proposed solution. The combination of known algorithms in one way or another can be an interesting and new solution, but in this case, I do not find anything like that. Moreover, the review of similar solutions is quite detailed but limited in quantity, and the selected articles are old. Finally, a technical note - I have doubts whether all of the references have been used in the text, e.g., item 44.

Round 2

Reviewer 3 Report

Thanks to the authors for their explanations. In my opinion, the article has improved in quality. However, I still believe that the novelty of the proposed approach is very modest and that the readability due to the very long text is poor. In this sense, the authors' answers did not convince me.
Besides, in the new text, you can notice linguistic errors (eg, A.E.S. in line 286, AEs in line 1031, python in Appendix Table A1) or the lack of math mode in formulas in lines 985-997.
As for other substantive issues, I have a note on the speed of the proposed approach. The authors show results for very small files. How does this work for actual size files, such as 1MB, 10MB, or 100MB? How will the steganography stage deal with such data?
